# Modelling the Effects of Beverage Substitution during Adolescence on Later Obesity Outcomes in Early Adulthood: Results from the Raine Study

**DOI:** 10.3390/nu11122928

**Published:** 2019-12-03

**Authors:** Miaobing Zheng, Anna Rangan, Rae-Chi Huang, Lawrence Joseph Beilin, Trevor Anthony Mori, Wendy Hazel Oddy, Gina Leslie Ambrosini

**Affiliations:** 1Institute for Physical Activity and Nutrition, School of Exercise and Nutrition Sciences, Deakin University, Geelong, Victoria 3125, Australia; 2School of Life and Environmental Sciences, The University of Sydney, Sydney 2006, New South Wales, Australia; anna.rangan@sydney.edu.au; 3Telethon Kids Institute, University of Western Australia, Perth 6009, Western Australia, Australia; Rae-Chi.Huang@telethonkids.org.au (R.-C.H.); Gina.Ambrosini@health.wa.gov.au (G.L.A.); 4Medical School, Royal Perth Hospital Unit, University of Western Australia, Perth 6009, Western Australia, Australia; lawrie.beilin@uwa.edu.au (L.J.B.); trevor.mori@uwa.edu.au (T.A.M.); 5Menzies Institute for Medical Research, University of Tasmania, Hobart 7005, Tasmania; wendy.oddy@utas.edu.au; 6School of Population and Global Health, University of Western Australia, Perth 6009, Western Australia, Australia

**Keywords:** sugar-sweetened beverages, obesity, waist circumference, substitution modelling

## Abstract

High sugar-sweetened beverage (SSB) consumption has been linked with obesity. The present study examined the associations between adolescent SSB intake and body mass index (BMI), waist circumference (WC), and overweight status in early adulthood, and modelled the association of alternative beverage substitution with BMI and WC. Data of offspring from the Western Australian Pregnancy Cohort (Raine) Study at ages 14 and 22 years were used (*n* = 667). SSB intake at 14 years (100 g/day) was associated with higher BMI (*β* = 0.19 kg/m^2^, 95% CI 0.04, 0.33), WC (*β* = 0.41cm, 95% CI 0.04, 0.78), and being overweight at 22 years (OR 1.10, 95% CI 1.02, 1.18). Every 100g modelled substitution of SSB with milk at age 14 years was associated with lower BMI (−0.19 kg/m^2^) and WC (−0.52 cm) at age 22 years. Replacement of SSB with diet drink was associated with higher BMI and WC. No association was found for substitutions of SSB with water, tea/coffee, or 100% fruit juice with BMI or WC. SSB intake during adolescence was associated with higher BMI, WC, and being overweight in early adulthood. Milk as an alternative to SSB was associated with less adiposity. Caution is necessary in recommending diet drinks as a SSB alternative.

## 1. Introduction

Elevated overweight and obesity prevalence and its concomitant adverse health outcomes have sparked widespread global health attention [1]. A transition towards high consumption of energy-dense and nutrient-poor foods, resulting in excess energy intake, is suggested as a primary underlying contributor to obesity [2]. Increased consumption of sugar-sweetened beverages (SSBs) over the past decades is of particular concern [3].

The association between SSB consumption and obesity has been widely studied [4]. The World Health Organisation concluded that consumption of food and beverages high in added sugar is a contributor to the prevalence of obesity [5,6]. Several recent reviews also concluded a positive association between SSB consumption and obesity, however, the underlying mechanism remains unclear [7,8,9]. Several hypotheses have been proposed to link SSB consumption with obesity. The most prevailing one is the poor compensation of energy consumed in liquid form, leading to excess energy intake and subsequent weight gain in the long-term [10]. Furthermore, energy intake may be a mediating factor underlying the association between SSB consumption and obesity [9]. Other hypotheses are that high glycaemic load of SSB exacerbates insulin response and increases the obesity risk, and hepatic metabolism of fructose promotes body weight and fat gain through de novo lipogenesis [8]. Moreover, SSB intake may be a marker of poor dietary patterns that contribute to obesity [10].

The majority of studies examining the association between SSB consumption and later obesity were conducted among children with short follow-up periods within childhood or adolescence, and did not assess dose-response relationships [8,11,12]. Only a few studies have evaluated the long-term association across life-stages from adolescence to adulthood [13,14,15,16], and only one of them assessed the potential dose–response relationship [16]. The transition from adolescence to adulthood involves marked physiological changes, and whether SSB consumption remains a potent risk factor for obesity during this period is yet to be confirmed.

A recent review revealed that there is a paucity of studies assessing the effects of SSB substitution with alternative beverages on later obesity outcomes [17]. The totality of evidence from randomised controlled trials (RCTs) and prospective cohort modelling studies suggest that substitutions of SSB with beverages that are lower in energy and higher in nutrients such as water, tea, coffee, and milk were found to be inversely associated with a range of obesity outcomes among both children and adults [17]. However, most RCTs examined short term effects of SSB substitution and study compliance was poor. Prospective cohort modelling studies managed to simulate the long-term substitution effects, but they were conducted in either children or adults. To best of our knowledge, no study has investigated the long-term effects of SSB substitution in adolescence and subsequent obesity outcomes in adulthood. Given that adolescence and early adulthood represent the periods of highest SSB consumption among the Australian population, the effects of SSB substitution on obesity outcomes during this period may be more prominent. Moreover, as considerable fat deposition occurs during progression to young adulthood, late adolescence provides a unique and responsive time for obesity prevention and management [18]. As a result, the investigation of the potential modifiable determinants of obesity from adolescence to early childhood is warranted.

This study aimed to investigate the association of adolescent SSB intake at age 14 years, in relation to body mass index (BMI), waist circumference (WC), and overweight status in early adulthood at age 22 years using data from a prospective Australian cohort. By using the statistical technique of substitution modelling, we sought to model the association between SSB substitution during adolescence and BMI and WC in early adulthood.

## 2. Materials and Methods 

### 2.1. Subjects

The Western Australian Pregnancy Cohort (Raine) Study is a cohort study that recruited 2900 pregnant women at 16–20 weeks gestation through antenatal clinics in Perth, Western Australia, between 1989 and 1991. Of the cohort, 2804 women had 2868 live births. These children were followed at regular intervals until early adulthood and are referred to as the Raine Study Generation 2 [19]. Data of these children at 14 and 22 years of age were used. At age 14 years, parental and sociodemographic factors and dietary intake were collected through questionnaires. Physical assessment at age 14 years was undertaken at the Telethon Kids Institute, Perth, Western Australia [19]. At age 22 years, the physical assessment was conducted at the Raine Study house on the University of Western Australia campus [20]. All data collection for the Raine Study occurred in accordance with the Australian National Health and Medical Research Council Guidelines for Ethical Conduct in Human Research. All subjects gave their informed consent for inclusion before they participated in the study. The study was conducted in accordance with the Declaration of Helsinki. The 22 year follow-up was approved by the Human Research Ethics Committees at the University of Western Australia (RA/4/1/5202) and Curtin University (HR67/2013). A detailed study protocol has been reported previously [20,21].

### 2.2. Anthropometrics

Study participants attended the study clinic for physical assessments at both ages 14 and 22 years. Trained personnel used standardised protocols to measure anthropometrics [20]. Body weight and height were measured using electronic scales in light clothing and a stadiometer, respectively [20,21]. Body mass index (BMI) was calculated as body weight divided by height squared in kilograms per metres squared (kg/m^2^). Waist circumference (WC) was measured by metal tape measure by trained observers at the level of the umbilicus to the nearest 0.1 cm. All anthropometric variables were measured twice, and the average of the two measurements was used. There were various cut-offs used for classifying overweight and obesity in children. In line with the recommendation for population studies, the International Obesity Task Force age- and sex-specific BMI cut-offs for children aged 2 to 18 years were used to classify overweight and obesity at age 14 years [22]. Similarly, BMI is a widely used screening tool for overweight and obesity for adults aged 19 years and over. Overweight or obesity at age 22 years were defined as BMI ≥ 25 kg/m^2^.

### 2.3. Dietary and Beverage Intake

Dietary intake at age 14 years was assessed by a semi-quantitative parent-assisted food-frequency questionnaire (FFQ) developed by the Commonwealth Scientific and Industrial Research Organization (CSIRO) in Adelaide, Australia [23]. The FFQ estimates the usual dietary intake over the previous year, and has been validated in this cohort against 3 day weighed food record with correlation coefficients of nutrient intakes between the two methods ranging from 0.11 to 0.52 [24].

Participants were asked to report the frequency of consumption (never, rarely, number of times per month, number of times per week, and number of times per day) and usual portion size of 212 food and beverage items. Frequency of consumption of each FFQ item was converted into daily equivalents and multiplied by the reported individual’s usual portion size to derive daily consumption. Detailed dietary assessment was reported previously [25,26]. Six beverage types were evaluated in the present study: (1) sugar-sweetened beverages (carbonated soft drinks including cola, cordials or fruit drink concentrate, and fruit juice drinks with the exclusion of 100% fruit juice); (2) plain water (spring and mineral water); (3) tea and coffee (plain and sweetened); (4) diet drinks (low calorie, artificially sweetened drinks); (5) 100% fruit juice (100% fruit and vegetable juices); and (6) milk (whole, reduced fat, skim, dairy, and soy milk).

### 2.4. Covariates

Dietary misreporting was evaluated by the Goldberg cut-off method [27]. The 95% confidence limits of the ratio of reported energy intake to basal metabolic rate was calculated to identify energy intake under-reporters and over-reporters. Three questions relating to participants’ physical activity level at school and outside school hours were used to estimate physical activity level: (1) time spent exercising vigorously during physical education at school (none, about a quarter of the time, about half the time, more than half the time, to almost all the time—with scores ranging from 0 to 4); (2) how often they got out of breath or sweated while exercising outside school hours (once a month, once a week, 2–3 times a week, 4–6 times a week, every day—with scores ranging from 0 to 4); and (3) how many hours they got out of breath or sweated while exercising outside school hours (none, about 1/2 hours per week, about 1 hour per week, about 2–3 hours per week, about 4–6 hours per week, 7 or more hours per week—with scores ranging from 0 to 5 ). Responses of these three questions were summed to create a composite score with a maximum score of 13. Consistent with our previous analysis, participants in the lowest tertile were classified as “low active” and the other tertiles were classified as “active” [28]. Maternal education level and family income were obtained from the parent questionnaire to represent the socioeconomic status of participants. Maternal education was categorised as high (≥10 years of education) or low (<10 years of education) [26]. Family income at age 14 years was categorised into quartiles. As the relationship between SSB intake and poor dietary patterns (measured as z-scores) has been widely acknowledged, previously identified dietary patterns in this cohort were included as potential covariates [25]. These included a western dietary pattern (high intake of fat, saturated fat, cholesterol, and refined sugars) and a healthy dietary pattern (high intakes of fibre and micronutrients) [23].

### 2.5. Statistical Methods

SSB intake was analysed as both continuous and categorical variables to explore a potential dose-response relationship. SSB intake was categorised into three groups: non-consumers, ≤1 serving per day, and >1 serving per day (one serving was equivalent to 1 cup (250 mL or 261 g)) [26]. This categorisation was chosen for ease of interpretation and comparison with most of the existing literature that used SSB servings to examine dose-response relationships. Descriptive analysis was conducted to explore baseline characteristics according to SSB intake categories. Difference of baseline characteristics among SSB intake categories was tested by analysis of variance or chi-square test. Multivariable linear regression models were used to examine associations between SSB intake at age 14 years and BMI/WC at age 22 years. The model was adjusted for baseline BMI/WC, intakes of water, tea and coffee, diet drink, 100% fruit juice, milk, and age, sex, dietary misreporting, physical activity, maternal education, family income, and both healthy and western dietary pattern scores. Logistic regression was conducted to assess the association between SSB intake at age 14 years and odds of being overweight or obese at age 22 years. The model was adjusted for overweight status at baseline and aforementioned covariates as in the linear regression analyses. Trend analysis among SSB intake categories was conducted by modelling SSB categories as a continuous variable. To evaluate whether total energy intake has a mediating effect on the association, additional adjustment for total energy intake was performed.

Substitution models were conducted to assess the associations between substitutions of SSB with other beverage types at 14 years on BMI and WC at 22 years [29,30]. The substitution model included total beverage intake and intakes of individual beverage alternatives for SSBs (plain water, tea and coffee, diet drink, 100% fruit juice, and milk), whereas SSB intake as the reference category is left out of the model.

BMI/WC = β1 × total beverage + β2 × plain water + β3 × tea and coffee + β4 × diet drinks + β5 × fruit juice + β6 × milk

Substitution model, where β1 to β6 are regression coefficients

When intakes of total beverage, tea and coffee, diet drink, 100% fruit juice, and milk are held constant, an increase in water intake results in a corresponding decrease in SSB intake. The coefficient of plain water (β2) indicates the effect of substituting water (100 g/day) for the same amount of SSBs on BMI and WC. Similarly, the coefficients of tea and coffee, diet drinks, 100% fruit juice, or milk have the same substitution meaning [29]. The substitution model was also adjusted for the aforementioned covariates. Another widely used substitution method is including SSB and the alternative beverages simultaneously in the same model, where the estimates for substitution association are obtained by the difference between beta-coefficient for SSB and the alternative beverage, and the variances or 95% confidence intervals of the substitution estimates are obtained from the variance–covariance matrix. These two approaches are statistically equivalent and will provide the same results [29]. The substitution model that provides direct substitution effects was used in the present analyses.

Sensitivity analysis excluding energy misreporters was conducted to explore the effect of dietary misreporting on the association. All analyses were conducted in SPSS 24.0 (SPSS Inc., Chicago, IL, USA) and a statistical significance of 0.05 was considered (two-sided).

## 3. Results

Complete data on dietary intake, anthropometric, and covariates at age 14 years were available for 1164 participants. At age 22 years (mean ± SD: 22.2 ± 0.6 years), 667 participants had anthropometric measures and were included in the current analyses. Compared to participants, non-participants had higher BMI and WC, higher intakes of diet drinks, SSB, total energy intake, and the western dietary pattern, as well as lower intake of water and lower healthy dietary pattern scores. Moreover, non-participants had a greater proportion of low-educated mothers and low family income earners (first two quartiles). All other variables were not significantly different (Appendix A). Baseline beverage intake at age 14 years (mean ± SD: 14.0 ± 0.2 years) is presented in Appendix A.

Baseline characteristics of the cohort according to SSB intake are shown in Table 1. Subjects who consumed >1 serving of SSB (mean intake 527 g/day) had significantly higher BMI and WC than those who consumed ≤1 serving of SSB (mean intake 116 g/day). Prevalence of overweight or obesity was higher among adolescents who consumed >1 serving of SSB than those who consumed ≤1 serving of SSB. Intakes of tea and coffee, 100% fruit juice, and milk were similar among SSB intake categories. However, higher SSB intake was associated with lower water and diet drink intakes, lower healthy dietary pattern z-scores, higher total energy intake, and western dietary pattern z-scores. SSB consumers had a lower proportion of highly educated mothers and a greater proportion had a lower family income compared to non-consumers.

Linear and logistic regression analyses of SSB intake at age 14 years and BMI and WC odds of being overweight or obese at age 22 years are presented in Table 2. After adjusting for covariates, every 100 g increase in SSB intake was associated with 0.19 kg/m^2^ and 0.41 cm higher BMI and WC, respectively at age 22 years (model 1). When SSB intake was modelled as a categorical variable, relative to SSB non-consumers, those consuming >1 serving per day of SSB at age 14 years had significantly higher BMI (*β* = 2.0 kg/m^2^, *P* = 0.02) and WC (4.47 cm, *P* = 0.03) at age 22 years (*P*_trend_ < 0.001) (Model 1). SSB intake (100 g/d) at 14 years was associated with higher odds of being overweight or obese (odds ratio (OR) 1.10, 95% CI 1.02, 1.18) at age 22 years. Adolescents who consumed >1 serving of SSB per day had 87% greater odds of being overweight or obese at age 22 years compared to non-consumers (*P*_trend_ = 0.01). Adjustment for total energy intake had little effects on these associations (model 2). Sensitivity analysis excluding misreporters revealed that direction of the association between SSB and adiposity outcomes remained the same, but the association became statistically non-significant.

Figure 1 illustrates regression analysis results for associations between substitution of SSB by plain water, tea and coffee, diet drinks, 100% fruit juice, and milk at age 14 years, and BMI and WC at age 22 years. After adjusting for all covariates, every 100g/day substitution of SSB with milk at age 14 years was associated with lower BMI (−0.19 kg/m^2^) and lower WC (0.52 cm) (*P* < 0.02) at age 22 years, respectively. However, 100g/day replacement of SSB with diet drinks was associated with higher BMI (1.29 kg/m^2^) and higher in WC (2.95 cm) (*P* < 0.05). No significant association was found between substitutions of SSB with 100% fruit juice, tea and coffee, or plain water after adjusting for all covariates (*P* > 0.05).

## 4. Discussion

In an Australian prospective cohort, we found positive associations between adolescent SSB intake and BMI, WC, and overweight status in early adulthood. Every 100 g higher SSB intake at age 14 years was associated with 0.19 kg/m^2^ and 0.42 cm increase in BMI and WC at age 22 years, respectively. Higher SSB intake at age 14 years was also associated with higher odds of being overweight and obese at age 22 years. When SSB intake was modelled as a categorical variable, a linear dose–response relationship was found (*P*_trend_ ≤ 0.01). Moreover, substitution of SSB with milk at age 14 years was associated with lower BMI and WC at age 22 years, whereas replacing SSB with diet drinks was associated with higher BMI and WC.

Few studies have examined the association between SSB intake and later obesity outcomes with long-term follow-up. Consistent with our findings, Viner et al. found SSB intake at age 16 years was associated with higher BMI 14 years later in adulthood in a British cohort [16]. In a longitudinal 21 year follow-up study of Finnish children and adolescents (aged 3–18 years), higher SSB consumption was associated with an increase in BMI among females [13]. Similar findings were also found in two U.S. cohorts from childhood to adolescence with 10 years of follow-up [14,31]. However, no association was observed in a cohort of Danish children between SSB intake at age 9 years and BMI, WC, or sum of four skinfolds at age 21 years [15]. The authors suggest that changes in dietary habits and being unable to control for a range of confounding factors may have contributed to their null findings.

Our study found a linear dose–response relationship between SSB intake and BMI and WC when SSB intake was analysed as a continuous or a categorical variable. Several studies have evaluated the linear dose–response relationship between SSB intake and obesity outcomes with mixed findings. Significant linear trends were found between SSB intake and a number of obesity outcomes including BMI [32,33,34,35], WC [35], percentage body fat [34,36], and obesity risk [33] in cohorts of children and adolescents from the U.S. [32,33], Australia [34,36], Colombia [35], and Denmark [37]. In contrast, some studies reported threshold effects of SSB intake on obesity outcomes [16,31,38,39]. The discrepancies may be due to imprecision in dietary assessment, as well as differences in underlying dietary patterns. Our study found adjustment for total energy intake had little effects on the association between SSB intake and BMI or WC, indicating that total energy intake may not be a mediating factor. It is likely that the non-energy effect of SSB intake such as glycaemic load and/or hepatic metabolism of fructose also contributed to the association. This has also been reported in other cohorts of Australian and Danish children [15,34].

Few studies examined effects of replacing SSB with milk on obesity outcomes and revealed mixed findings. Aligning with our finding, milk as an alternative for SSB demonstrating beneficial effects on long-term obesity outcomes was also reported in two other studies of Danish children [37,40]. It is hypothesised that several substances contained in milk may have positive effects on body weight and fat regulation. Calcium impacts adipocyte lipid metabolism, fat oxidation, and fatty acid absorption, mechanisms that can suppress fat gain/storage and prevent body weight gain [41,42]. Moreover, milk proteins such as whey protein may enhance satiety and reduce food intake [43]. Other than the direct effect of the milk intake per se, another hypothesis is that milk intake is a marker of a healthier lifestyle. The milk category in the present analysis included mainly plain dairy milk. Other milk alternatives (i.e., soy) although included, had a low intake. It is worth noting that despite the fact that milk alternatives and flavoured milk are rich in calcium, they are often high in added sugar and may not be good alternatives for SSB. Diet drinks are proposed as a low energy alternative to SSBs. Controversy remains with regards to diet drinks and obesity [44]. Data from animal studies associated diet drink intake with increased food intake, and body weight and fat gain when compared with SSB [45]. However, evidence from human studies suggest no confirmative evidence to ascertain diet drinks as a cause of obesity [46]. Findings from intervention trials revealed a probable beneficial effect of replacing SSBs with diet drinks on obesity [46]. Our findings that SSB, when replaced with diet drinks, was associated with higher BMI and WC should be interpreted with caution. It should be noted that the overall intake of diet drinks was low, and SSB consumers had the lowest diet drink intake. Given our observational design, reverse causation in overweight and obese subjects consuming diet drinks to lose weight is possible, as intake was assessed at only one time-point.

Substitution of SSB with 100% fruit juice was not associated with BMI or WC. Despite the fact that 100% fruit juice contains more nutrients than SSB, it is not a suitable alternative for SSB due to the high sugar content and lack of fibre. The American Academy of Paediatrics suggests excess fruit juice intake may contribute to increased energy intake and weight gain [47]. Dietary guidelines also recommend that 100% fruit juice should be limited to prevent weight gain and dental caries in children [48,49]. As a natural kilojoule-free beverage, water as an alternative for SSB theoretically would result in a reduction of energy intake, and potentially weight loss. However, limited studies have evaluated water intake per se with obesity, and the totality of evidence remains inconclusive [50,51]. The lack of an association in our study was likely attributable to underreporting of water intake because tap water as a major source of water consumption was not measured in our study. Likewise, tea and coffee in their basic form are low kilojoule alternatives for SSB, and the catechins and caffeine may have potential weight reduction effects [52,53]. Despite this, we found no beneficial effects of tea/coffee on body weight when substituted for SSB. Low consumption of tea/coffee in this adolescent population and inclusion of both plain and sweetened tea and coffee may contribute to lack of any observed effect.

Our study has several strengths. It is the first study to examine the prospective association of SSB intake with BMI, WC, and overweight status from adolescence to early adulthood among an Australian prospective cohort. We also explored whether there was a linear dose–response relationship as well as the mediating effect of total energy intake on the association. Furthermore, we simulated the effects of SSB substitution with other beverages on long-term changes in BMI and WC using substitution modelling. Other strengths include a long follow-up period and examination of a wide range of important covariates such as socioeconomic status, physical activity, and dietary patterns. The availability of dietary pattern scores is valuable, as SSB intake may be interrelated with other dietary factors contributing to obesity. Limitations include the observational nature of our study, which means that causal relationships cannot be determined. Beverage intake was self-reported by participants through a validated FFQ, and misreporting cannot be dismissed [54]. The FFQ was validated in this population [24], and we did assess dietary misreporting and considered this in our analysis. The loss to follow-up is another limitation of our study, and this will limit the generalisability of our results [19]. However, given the high proportion of drop-outs, it is not plausible to conduct missing data imputation [55]. We found non-participants had higher SSB intake, BMI, and WC than participants. It is conceivable that the association of SSB intake and BMI/WC may have been stronger if non-participants were included. Lastly, it is important to acknowledge that obesity is multifactorial and SSB intake is only one of the contributing factors.

## 5. Conclusions

In conclusion, our study has shown that higher SSB intake during adolescence is associated with higher BMI, WC, and greater odds of being overweight or obese in early adulthood. Modelled substitution of SSB with milk was associated with lower BMI and WC. However, further studies are warranted to investigate the long-term effects of other beverage types as alternatives to SSB for obesity prevention. Given the high consumption of SSB and concurrent high prevalence of obesity among children and adolescents, our findings contribute further evidence to support national guidelines, food polices, and public health campaigns to reduce SSB consumption. Moreover, our study indicates that SSB substitution with milk is likely to be an effective intervention strategy for obesity prevention and management among adolescents.

## Figures and Tables

**Figure 1 nutrients-11-02928-f001:**
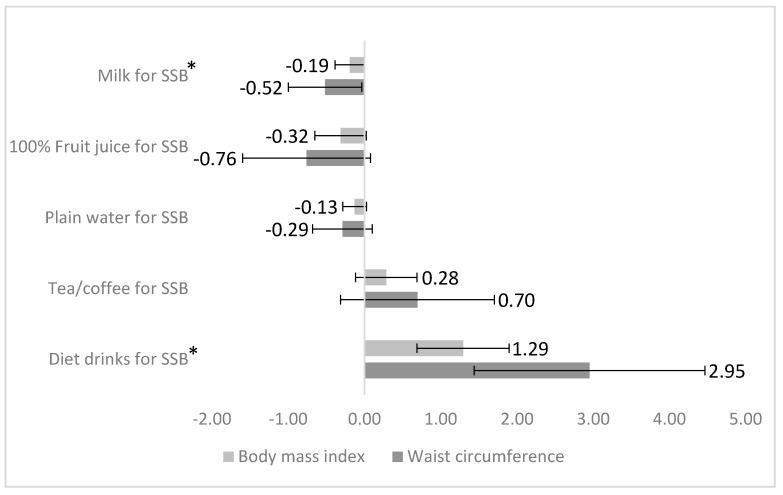
Predicted changes in body mass index (kg/m^2^) and waist circumference (cm) (error bars indicate 95% confidence interval) at age 22 years associated with 100 g/day substitution of sugar-sweetened beverages (SSB) with plain water, tea/coffee, diet drink, 100% fruit juice, and milk. Substitution model included plain water, tea/coffee, diet drinks, 100% fruit juice, and milk (100 g/day). SSB is excluded from the model (reference category). By keeping the total beverage intake constant, a unit increase in any single beverage intake represents a corresponding decrease in SSB intake. Model adjusted for age, sex, BMI, WC, dietary misreporting, physical activity, maternal education, family income, healthy dietary pattern, and western dietary pattern scores at baseline (age 14 years). Asterisk (*) indicates statistical significance (*P* < 0.05).

**Table 1 nutrients-11-02928-t001:** Baseline study characteristics at age 14 years according to sugar-sweetened beverage (SSB) intake *.

	Sugar-Sweetened Beverage Intake					
	Non-Consumer (*n* = 40)		≤1 Serving (*n* = 363)		>1 Serving (*n* = 264)	
Continuous variables	Mean	SD	Mean	SD	Mean	SD
Age (years)	14.0	0.2	14.0	0.2	14.0	0.2
† Body mass index (kg/m^2^)	21.1	3.9	20.6	3.4	21.7	4.4
† Waist circumference (cm)	74.1	10.0	73.6	9.2	76.7	11.5
† Plain water (g)	785.2	608.4	809.8	521.9	649.2	545.2
Tea/coffee (g)	38.1	84.7	46.7	109.1	41.0	78.3
† Diet drinks (g)	50.1	122.0	15.0	48.6	17.8	63.3
100% fruit juice (g)	89.5	118.1	92.1	117.6	89.4	132.5
Milk (g)	398.9	332.1	398.2	320.5	387.3	311.1
† Total energy intake (MJ)	8.4	2.2	8.9	2.7	10.3	3.1
† Healthy dietary pattern z-score	0.3	0.8	0.1	0.8	−0.1	0.8
† Western dietary pattern z-score	−0.7	0.5	−0.3	0.7	0.3	0.9
Categorical variables	*n* (%)		*n* (%)		*n* (%)	
† Overweight or obese	5 (12.5)		34 (9.4)		48 (18.2)	
Sex						
Males	20 (50.0)		175 (48.2)		123 (46.6)	
Females	20 (50.0)		188 (51.8)		141 (53.4)	
† Dietary misreporting						
Under-reporters	14 (35.0)		107 (29.5)		59 (22.3)	
Plausible reporters	25 (62.5)		229 (63.1)		162 (61.4)	
Over-reporters	1 (2.5)		27 (7.4)		43 (16.3)	
Physical activity						
Low active	10 (25.0)		158 (43.5)		110 (41.8)	
Active	30 (75.0)		205 (56.5)		154 (58.2)	
† Maternal education						
High	33 (82.5)		267 (74.1)		164 (62.1)	
Low	7 (17.5)		94 (25.9)		100 (37.9)	
† Family income (rank)						
0	9 (22.5)		77 (21.2)		78 (29.5)	
1	13 (32.5)		95 (26.2)		89 (33.7)	
2	5 (12.5)		98 (27.0)		56 (21.2)	
3	13 (32.5)		93 (25.6)		41 (15.5)	

* SD: standard deviation. One serving of sugar-sweetened beverage was equivalent to 1 cup (250 mL or 261 g). SSB intake (mean ± SD): ≤1 serve (115.9 ± 78.5) and >1 serve (526.8 ± 289.1); family income: 0 is lowest and 3 is highest. † Difference of baseline characteristics among SSB consumption categories tested by analysis of variance or chi-square tests: *P*-value < 0.05.

**Table 2 nutrients-11-02928-t002:** Multivariable regression analysis between sugar-sweetened beverage intake at age 14 years and body mass index (BMI, kg/m^2^) and waist circumference (WC, cm) and overweight or obesity at age 22 years. *

	BMI			WC			Overweight or Obesity		
	*β*	95% CI	*P*	*β*	95% CI	*P*	OR	95% CI	*P*
Continuous variable (100 g/day)									
Crude model	0.21	(0.08, 0.33)	0.001	0.41	(0.09, 0.73)	0.01	1.08	(1.02, 1.14)	0.01
Model 1	0.19	(0.04, 0.33)	0.01	0.41	(0.04, 0.78)	0.03	1.10	(1.02, 1.18)	0.01
Model 2	0.16	(0.01, 0.30)	0.04	0.34	(−0.04, 0.71)	0.08	1.10	(1.02, 1.18)	0.01
Categorical variable									
Crude									
Non-consumer	ref								
≤1 serve	−0.18	(−1.72, 1.37)	0.82	−1.34	(−5.27, 2.60)	0.51	0.92	(0.43, 1.98)	0.83
>1 serve	1.69	(1.21, 3.25)	0.04	2.68	(−1.32, 6.67)	0.19	1.77	(0.81, 3.84)	0.15
Model 1									
Non-consumer	ref								
≤1 serve	0.32	(−1.27, 1.90)	0.70	0.39	(−3.49, 4.27)	0.84	0.96	(0.41, 2.25)	0.92
>1 serve	2.00	(0.32, 3.67)	0.02	4.47	(0.35, 8.58)	0.03	1.87	(0.76, 4.62)	0.18
Model 2									
Non-consumer	ref								
≤1 serve	0.41	(−1.17, 2.00)	0.61	0.35	(−3.56, 4.26)	0.86	0.93	(0.41, 2.25)	0.93
>1 serve	1.98	(0.31, 3.64)	0.02	4.09	(−0.04, 8.22)	0.05	1.84	(0.75, 4.75)	0.19

* *β*: regression coefficient; CI: confidence interval, OR: odds ratio. Model 1: the model adjusted for baseline BMI, WC, overweight or obesity and other covariates including intakes of water, tea/coffee, diet drink, 100% fruit juice, and milk, age, gender, dietary misreporting, physical activity, maternal education, family income, healthy dietary pattern, and western dietary pattern scores at age 14 years. Model 2: additional adjustment for total energy intake upon model 1. * Crude: BMI *P*-trend < 0.0001, WC *P*-trend = 0.001, overweight or obesity *P*-trend = 0.001. * Model 1: BMI *P*-trend < 0.0001, WC *P*-trend < 0.0001, overweight or obesity *P*-trend = 0.002. * Model 2: BMI *P*-trend = 0.001, WC *P*-trend < 0.0001, overweight or obesity *P*-trend = 0.003.

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
