# Peer review of "Modelling the Effects of Beverage Substitution during Adolescence on Later Obesity Outcomes in Early Adulthood: Results from the Raine Study"

_nutrients, 2019, doi:10.3390/nu11122928_

Round 1

Reviewer 1 Report

The work is very interesting, it covers a long study time and a sufficient group of respondents.

Few studies have examined the association between SSB intake and later obesity outcomes with long-term follow-up, but no work shows such a long observation period. An interesting aspect would also be the connection with the birth weight of the examined people and the possible occurrence of macrosomia or hypotrophy in the neonatal period.
This may be a topic for the next work in connection with the type of feeding in 1 year of life and how to expand the diet, if the authors have such data. On the other hand, it is known that not only the consumption of beverages, including SSB, is a factor in the development of overweight and obesity.
In my opinion the authors should better present the method of anthropometric measurements both at the age of 14 and 22, who took them and where. Similarly, the scope of overweight and obesity recognition in both age groups should be presented in detail.

Author Response

The work is very interesting, it covers a long study time and a sufficient group of respondents.

Few studies have examined the association between SSB intake and later obesity outcomes with long-term follow-up, but no work shows such a long observation period. An interesting aspect would also be the connection with the birth weight of the examined people and the possible occurrence of macrosomia or hypotrophy in the neonatal period.
This may be a topic for the next work in connection with the type of feeding in 1 year of life and how to expand the diet, if the authors have such data. On the other hand, it is known that not only the consumption of beverages, including SSB, is a factor in the development of overweight and obesity.

Reply: Thank you for your time in reviewing our paper and suggestions. Your advice to look at how birth weight and the early period of feeding on the development of obesity is very important and interesting to explore. Our colleagues have previously investigated the influence of infant feeding (both breastfeeding and formula introduction) and development of obesity from ages 1 to 20 years with adjustment for birth weight in the same cohort. Please see the details of the publication: https://www.ncbi.nlm.nih.gov/pubmed/25300269

We acknowledge that obesity is multifactorial, and it is a developmental process attributable to a complex interaction of factors along the life course. SSB consumption is indeed not the only contributing factor in the obesity epidemic. We have now highlighted this in our limitation section in line 343: “Lastly, it is important to acknowledge that obesity is multifactorial and SSB intake is only one of the contributing factors.”

In my opinion the authors should better present the method of anthropometric measurements both at the age of 14 and 22, who took them and where. Similarly, the scope of overweight and obesity recognition in both age groups should be presented in detail.

Reply: Study participants attended the study clinic for physical assessments at both ages 14 and 22 years. Trained personnel used standardised protocols to measure anthropometrics (now added to line 102-103). The following text has been added to provide some background on the classification of overweight and obesity in both children and adults used in this analysis: “There are various cut-offs used for classifying overweight and obesity in children. In line with the recommendation for population studies, the International Obesity Task Force age- and sex-specific BMI cut-offs for children aged 2 to 18 years were used to classify overweight and obesity at age 14 years. Similarly, BMI is a widely used screening tool for overweight and obesity for adults aged 19 years and over. Overweight or obesity at age 22 years were defined as BMI≥25kg/m2.” (line 108-113)

Reviewer 2 Report

In the present work, authors Miaobing Zheng and collaborators analysed the impact of beverage substitution during adolescence on later obesity outcomes in early adulthood.

As far as high sugar-sweetened  beverage  (SSB)  consumption  implication in obesity development is concerned, this MS deals about a subject of great interest.

This study is based on data of  offspring  from  the  Western Australian Pregnancy Cohort (Raine) Study at ages 14 and 22 years. Authors used statistical analysis tools and modelled the association of  alternative  beverage  substitution  with  anthropometric values (body mass index BMIU, waist circumference WC ...) 

Authors found that substitution of SSB with milk at 14 years was associated with BMI and WC decreases at 22 years. Replacement of SSB with diet drinks was associated with higher BMI and WC whereas no association was found for substitutions of SSB with water, tea/coffee or fruit juice with BMI or WC.

The MS is very well written. I would like to address few minor points to Authors:

Page 4 lane 164, the formula is not very simple, what beta letter stand for? Table 1, I suggest the statistic sign to be put beside the data (and not the lane name) In table 1 please replace gender by sex. From figure 1 I understand no significant association was found for substitutions of SSB with fruit juice because of the big standard deviation. Authors could develop on this.

Author Response

In the present work, authors Miaobing Zheng and collaborators analysed the impact of beverage substitution during adolescence on later obesity outcomes in early adulthood.

As far as high sugar-sweetened  beverage  (SSB)  consumption  implication in obesity development is concerned, this MS deals about a subject of great interest.

This study is based on data of  offspring  from  the  Western Australian Pregnancy Cohort (Raine) Study at ages 14 and 22 years. Authors used statistical analysis tools and modelled the association of  alternative  beverage  substitution  with  anthropometric values (body mass index BMIU, waist circumference WC ...) 

Authors found that substitution of SSB with milk at 14 years was associated with BMI and WC decreases at 22 years. Replacement of SSB with diet drinks was associated with higher BMI and WC whereas no association was found for substitutions of SSB with water, tea/coffee or fruit juice with BMI or WC.

The MS is very well written. I would like to address few minor points to Authors:

Reply: Thank you for your time in reviewing our paper and suggestions.

Page 4 lane 164, the formula is not very simple, what beta letter stand for?

Reply: The beta letter stands for the regression coefficient representing the effect size of the association that is the effect of substituting 100g beverage alternative to 100g SSB on either BMI or WC. We have now added “where β1 to β6 are regression coefficients” next to the formula. (Line 183)

Table 1, I suggest the statistic sign to be put beside the data (and not the lane name) In table 1 please replace gender by sex. From figure 1 I understand no significant association was found for substitutions of SSB with fruit juice because of the big standard deviation. Authors could develop on this.

Reply: Given that we have three columns of data and for categorical variables with more rows, the table would look too cluttered if the sign was placed beside the data. To emphasise the statistical significant difference, we have now bolded the variable name. Please see Table 1. “Gender” in table 1 has now been replaced by “Sex”. The discussion on fruit juice as an alternative to SSB on body weight management is discussed in lines 320 to 324 (Substitution of SSB with 100% fruit juice was not associated with BMI or WC. Despite that 100% fruit juice contains more nutrients than SSB, it is not a suitable alternative for SSB due to the high sugar content and lack of fibre. The American Academy of Paediatrics suggests excess fruit juice intake may contribute to increased energy intake and weight gain[47]. Dietary guidelines also recommend that 100% fruit juice should be limited to prevent weight gain and dental caries in children [48,49]).

Reviewer 3 Report

Page 2, line 76 – you describe your study as a “large cohort” but the sample size is only 677 which is a fairly small sample to answer your research question, and would not be considered a large cohort.

Page 4, line 144 – I would change “serve” to “serving”

Table 1 – it appears as though many of the variables for beverages were highly skewed as the SD is almost as high (and sometimes higher) as the mean values. Did you test the variables for normality prior to further analysis? What impact did the inclusion of these variables have on the distribution of residuals from the linear models?

I am confused by what exactly you predicted – for example in the methods, and in line 204 you state that you predicted the delta BMI and WC (changes between 14 and 22 years), but in line 207-208 you show results for predicting BMI and WC at age 22 – not the changes. Also the results shown in the abstract are for BMI and WC at 22 years, not the change scores. One would think that if your primary analysis strategy is to analyze change scores, then you should put those primary results in the abstract. This is a major issue – the wording suggests that you are predicting BMI and WC at 22 years, but the analysis plan suggests it is change (i.e. cm per year in WC) that you are predicting.  

A weakness of the study is low follow-up from the original cohort size of 2900 births to the final sample of 677, or even the drop from 1164 at baseline to 667. Also to note – the abstract says 677 and the text results say 667 so there is a discrepancy there.

Table 2 - since these are changes in BMI and WC the units should be change in units per year.

Line 280 – another explanation for milk consumption being related to less weight gain is that the milk is a marker for an overall healthier lifestyle (and factors you haven’t measured) – rather than any direct effect of the milk per se. i.e. kids that drink more milk would be heathier overall compared to kids that drink more SSBs in general.

Line 311 – I am not convinced that your small sample size is a representative sample of Australia.

Author Response

Page 2, line 76 – you describe your study as a “large cohort” but the sample size is only 677 which is a fairly small sample to answer your research question, and would not be considered a large cohort.

Reply: Thank you for your time in reviewing our paper and providing valuable feedback. We have now removed ‘large’ from the sentence (line 80).

Page 4, line 144 – I would change “serve” to “serving”

Reply: This has now been changed (line 160).

Table 1 – it appears as though many of the variables for beverages were highly skewed as the SD is almost as high (and sometimes higher) as the mean values. Did you test the variables for normality prior to further analysis? What impact did the inclusion of these variables have on the distribution of residuals from the linear models?

Reply: Thank you for raising this important point. We would like to clarify some confusion around the assumptions of linear regression models regarding the normality of variables. Many researchers believe that multiple regression requires normality. This is not the case. Linear regression models make no assumptions about the distribution of the predictor variable. If this were the case than we would not be able to assess categorical or dummy coded variables as predictors in our models. Instead, assumption of normality is for residuals (the difference between the observed value of the dependent variable (y) and the predicted value (Å·) from the model). Moreover, normality of residuals is required for valid hypothesis testing, that is, the normality assumption assures that the p-values will be valid. Normality is not required in order to obtain unbiased estimates of the regression coefficients. Nevertheless, a highly skewed predictor variable may be made symmetric with a transformation or analysed as a categorical variable to further understand the linearity of the association. Please see some reference below for your information: https://stats.idre.ucla.edu/stata/webbooks/reg/chapter2/stata-webbooksregressionwith-statachapter-2-regression-diagnostics/ Due to the nature of dietary intake data, food intakes always tend to be skewed. We also did carefully consider the skewness of the predictor variables in our analysis. First, we did the test the residual of the model with these skewed beverage intake data, the residuals were normally distributed. We also did log-transform our beverage intake data and the results were all in the same direction. As log transformed beverage intake does not give us a readily interpretable effect size on the associations, the log-transformed results were not presented as the primary analysis. We did further test for linearity by analysing SSB intake as a categorical intake (presented in the table) and this further confirmed the linear relationship and the validity of our model.

I am confused by what exactly you predicted – for example in the methods, and in line 204 you state that you predicted the delta BMI and WC (changes between 14 and 22 years), but in line 207-208 you show results for predicting BMI and WC at age 22 – not the changes. Also the results shown in the abstract are for BMI and WC at 22 years, not the change scores. One would think that if your primary analysis strategy is to analyze change scores, then you should put those primary results in the abstract. This is a major issue – the wording suggests that you are predicting BMI and WC at 22 years, but the analysis plan suggests it is change (i.e. cm per year in WC) that you are predicting.  

Reply: Thanks for your comments and sorry for the confusion. The analyses predicting the association between beverage intake at 14 years and change in BMI/WC from ages 14 to 22 years (Model 1 below) or BMI/WC at 22 years (Model 2 below) when adjusted for BMI/WC at 14 years, give the same results as two models are statistically equivalent.  Model 1: ΔBMI/WC at 22 years=SSB intake at 14 years + BMI/WC at 14 years Model 2: BMI/WC at 22 years=SSB intake at 14 years + BMI/WC at 14 years Therefore, we used these wording interchangeably. However, to prevent confusion, we have now changed our wording to be consistent as predicting the association between SSB intake at 14 years and BMI/WC at 22 years.

A weakness of the study is low follow-up from the original cohort size of 2900 births to the final sample of 677, or even the drop from 1164 at baseline to 667. Also to note – the abstract says 677 and the text results say 667 so there is a discrepancy there.

Reply: Sorry for the typo, it should be 667 (line 23). Thank you for pointing this out, it has now been corrected. The loss to follow-up is another limitation of our study and this will limit the generalisability of our results. We also commented on how inclusion of non-participants would influence our results. Please see line 347-352 (The loss to follow-up is another limitation of our study and this will limit the generalisability of our results [19]. However, given the high proportion of drop-outs, it is not plausible to conduct missing data imputation [55]. We found non-participants had higher SSB intake, BMI and WC than participants. It is conceivable that the association of SSB intake and BMI/WC may have been stronger if non-participants were included).

Table 2 - since these are changes in BMI and WC the units should be change in units per year.

Reply: We have changed our wording as are modelling the effects of SSB intake (100g/day) at 14 years with BMI/WC at 22 years, therefore, the unit stay as kg/m2 and cm respectively.

Line 280 – another explanation for milk consumption being related to less weight gain is that the milk is a marker for an overall healthier lifestyle (and factors you haven’t measured) – rather than any direct effect of the milk per se. i.e. kids that drink more milk would be heathier overall compared to kids that drink more SSBs in general.

Reply: Thank you for that important point. This has been added to line 305-306: “Other than the direct effect of the milk intake per se, another hypothesis is that milk intake is a marker of a healthier lifestyle.”

Line 311 – I am not convinced that your small sample size is a representative sample of Australia.

Reply: In the original sampling of the Raine study, the representative of the sample has been matched with the overall Australian population. Yes, we agree, after loss of follow-up, the sample is likely to be non-representative. We have now replace “representative” with “prospective” in line 337.

Round 2

Reviewer 3 Report

I have no further comments.